# Insulin–Mimetic Dihydroxanthyletin-Type Coumarins from *Angelica decursiva* with Protein Tyrosine Phosphatase 1B and α-Glucosidase Inhibitory Activities and Docking Studies of Their Molecular Mechanisms

**DOI:** 10.3390/antiox10020292

**Published:** 2021-02-15

**Authors:** Md Yousof Ali, Susoma Jannat, Hyun Ah Jung, Jae Sue Choi

**Affiliations:** 1Department of Physiology and Pharmacology, Hotchkiss Brain Institute and Alberta Children’s Hospital Research Institute, Cumming School of Medicine, University of Calgary, Calgary, AB T2N 4N1, Canada; mdyousof.ali@ucalgary.ca; 2Department of Biochemistry and Molecular Biology, University of Calgary, Calgary, AB T2N 4N1, Canada; jannatacct@gmail.com; 3Department of Food Science and Human Nutrition, Jeonbuk National University, Jeonju 54896, Korea; jungha@jbnu.ac.kr; 4Department of Food and Life Science, Pukyong National University, Busan 48513, Korea

**Keywords:** *Angelica decursiva*, coumarins, PTP1B, α-glucosidase, glucose uptake, antioxidant

## Abstract

As a traditional medicine, *Angelica decursiva* has been used for the treatment of many diseases. The goal of this study was to evaluate the potential of four natural major dihydroxanthyletin-type coumarins—(+)-*trans*-decursidinol, Pd-C-I, Pd-C-II, and Pd-C-III—to inhibit the enzymes, protein tyrosine phosphatase 1B (PTP1B) and α-glucosidase. In the kinetic study of the PTP1B enzyme’s inhibition, we found that (+)-*trans*-decursidinol, Pd-C-I, and Pd-C-II led to competitive inhibition, while Pd-C-III displayed mixed-type inhibition. Moreover, (+)-*trans*-decursidinol exhibited competitive-type, and Pd-C-I and Pd-C-II mixed-type, while Pd-C-III showed non-competitive type inhibition of α-glucosidase. Docking simulations of these coumarins showed negative binding energies and a similar proximity to residues in the PTP1B and α-glucosidase binding pocket, which means they are closely connected and strongly binding with the active enzyme site. In addition, dihydroxanthyletin-type coumarins are up to 40 µM non-toxic in HepG2 cells and have substantially increased glucose uptake and decreased expression of PTP1B in insulin-resistant HepG2 cells. Further, coumarins inhibited ONOO^−^-mediated albumin nitration and scavenged peroxynitrite (ONOO^−^), and reactive oxygen species (ROS). Our overall findings showed that dihydroxanthyletin-type coumarins derived from *A*. *decursiva* is used as a dual inhibitor for enzymes, such as PTP1B and α-glucosidase, as well as for insulin susceptibility.

## 1. Introduction

Diabetes mellitus (DM) has risen quickly in recent decades and currently, over 415 million DM sufferers worldwide, predicted to hit 642 million by 2040 [1]. Moreover, the incidence of DM in younger people, even before puberty, is rising, including some obese children [2,3]. Type 2 diabetes mellitus (T2DM) is due to insulin resistance or deficiency, the predominant type of diabetes that accounts for over 90 percent of the DM incidences [3,4,5,6].

Protein tyrosine phosphatase 1B (PTP1B) is part of the intracellular PTP that is active in the negative regulation of insulin and leptin signaling systems [7]. PTP1B catalyzes the dephosphorylation of tyrosine residues in the activated insulin receptor β subunit (IR-β) and insulin receptor substrate-1 (IRS-1), resulting in decreased insulin regulation [8,9]. Additionally, PTP1B knockdown mice also exhibit improved insulin sensitivity for glucose and insulin tolerance testing, suggesting that insulin sensitivity is largely modulated by PTP1B [10,11]. PTP1B inhibitors are therefore potential therapeutic candidates for re-establishing insulin susceptibility and treating T2DM.

Reducing postprandial hyperglycemia and slowing glucose absorption by inhibiting carbohydrate-hydrolyzing enzymes, such as α-glucosidase, is one of the therapeutic approaches to restore insulin sensitivity [12]. α-Glucosidase plays a key role in transformation of oligosaccharides and disaccharides to glucose as an essential carbohydrate hydrolysis enzyme, and the monosaccharides formed can be absorbed by the small intestine and contribute to an increase in blood glucose levels [13,14,15]. Therefore, α-glucosidase has been recognized as the main target enzyme in preventing and treating T2DM.

In addition, oxidative stress has been shown to have a major impact on human DM and DM-related complications [16]. The key effects of oxidative stress in DM pathogenesis contribution to β-cell dysfunction and apoptosis, hyperglycemia, insulin resistance, hyperinsulinemia, and dyslipidemia [17,18]. There is growing evidence that prolonged exposure to high glucose, through glucose auto-oxidation and protein glycosylation, induces the development of free radicals, especially reactive oxygen species (ROS) [19]. The ROS cause of oxidative damage has been suggested to clarify the excess prevalence of DM vascular complications that can be mediated by oxidative stress [20,21]. In addition, nitrotyrosine is a product of ONOO^‒^ an action that can be indirectly inferred by nitrotyrosine residues in the production of ONOO^‒^ [22]. The role of nitrotyrosine as a possible risk factor in DM was recently highly considered and an increased level of nitrotyrosine in diabetic patients’ plasma was identified [23].

The genus *Angelica* L. belongs to the Apiaceae (*alt.* Umbelliferae) family, generally referred to as the parsley family, which includes over 60 essential biennial herbs species. Among the *Angelica* species, *Angelica decursiva* Fr. et Sav (Umbelliferae) is a perennial herb that is distributed widely in China, Japan, and Korea [24]. In traditional Chinese medicine (TCM), it is also used to treat diseases, such as cough caused by pathogenic wind heat and accumulation of phlegm and heat in the lungs [25]. *A. decursiva* is also used as a salad in Korea and is confirmed to be non-toxic [26]. The use of the *A. decursiva* roots in China have a long history of minimising fever, summer heat solving, and preventing bleeding. This plant was also used as an analgesic, antipyretic, antitussive, and cough treatment in traditional Korean medicine [27,28].

As a result, many classes of compounds have been isolated, including different types of coumarin derivatives: nodakenin; nodakenetin; isorutarine; umbelliferone; umbelliferone 6-carboxylic acid; Pd-C-III; 6-formyl umbelliferone; 2′-isopropyl psoralene; Pd-C-I; columbianadin; decursin; 4-hydroxy Pd-C-III; (+)-*trans*-decursidinol; Pd-C-II, and decursidin [27,28,29,30,31,32,33,34,35,36,37,38]. It has been stated that these compounds have a broad range of biological activities, including anti-hypertensive [36], anti-tumor [39], anti-platelet aggregation [40], neuroprotective [41], memory-enhancing [42], anti-amnesic [43], anti-inflammatory [27,28,44], anti-diabetic, anti-Alzheimer [29,30,31,32,33,34], and antioxidative [27,44], effects.

A recent study reported the potential anti-hypertensive, anti-diabetic, antioxidant, anti-inflammatory and anti-Alzheimer activities of MeOH extracts and its different solvent soluble fractions, and reported their constituents, as well [27,28,29]. In view of the reports, however, of the promising activities of *A. decursiva* and its constituents, no extensive research are available on the major four dihydroxanthyletin-type coumarins derived from *A. decursiva* and the possibility of developing T2DM drugs.

As a result, we examined the PTP1B and α-glucosidase inhibitory activities of the four major dihydroxanthyletin-type coumarins as part of our continuous research to find anti-diabetic agents in *A. decursiva*. In addition, to determine the type of enzymatic inhibition of PTP1B and α- glucosidase, enzyme kinetic analyses of these coumarins were carried out using Dixon and Lineweaver-Burk plots. The mechanisms of interactions between coumarins and binding sites in PTP1B and α-glucosidase, were also investigated in molecular modeling studies. At present, the mechanisms by which coumarins promote glucose uptake remain unclear; thus, the stimulatory effects of coumarins’ glucose uptake and the expression of PTP1B in insulin-resistant HepG2 cells are recorded herein. This is the first research to examine, to the best of our knowledge, the antidiabetic activity of these four coumarins and their underlying mechanisms.

## 2. Materials and Methods

### 2.1. General Experimental Procedures

In deuterated solvents (dimethylsulfoxide (DMSO)-*d*_6_, chloroform (CDCl_3_)), the ^1^H- and ^13^C-NMR spectra were acquired using a JEOL JNM ECP-400 spectrometer at 400 and 100 MHz, respectively. Column chromatography was conducted using sephadex LH20 (20–100 µm, Sigma, St. Louis, MO, USA), LiChroprep^®^ RP-18 (40–63 µm, Merck, Darmstadt, Germany), Diaion HP20 (250–850 µm, Sigma, St. Louis, MO, USA), and silica gel 60 (70–230 mesh, Merck, Darmstadt, Germany). On pre-coated Merck Kieselgel 60 F254 (20 × 20 cm, 0.25 mm, Merck, Darmstadt, Germany) and RP-18 F254s (5–10 cm, Merck, Darmstadt, Germany) plates, all TLC was carried out, using 50% H_2_SO_4_ as a spray reagent. 

### 2.2. Chemicals and Reagents

*p*-Nitrophenyl phosphate (*p*NPP), *p*-nitrophenyl α-D-glucopyranoside (*p*NPG), yeast α-glucosidase, trolox (>90% purity), acarbose, ursolic acid (>90% purity), metformin (>90% purity), L-penicillamine (>90% purity), phenylmethylsulfonyl fluoride (PMSF), bovine serum albumin (BSA), 3-(4,5-dimethylthiazol-2-yl)-2,5-diphenyl tetrazolium bromide (MTT), insulin from bovine pancreas, dimethyl sulfoxide (DMSO), and *tert*-butyl hydroperoxide (*t*-BHP) were purchased from Sigma-Aldrich Co. (St Louis, MO, USA). PTP1B (human recombinant) has been purchased from Biomol^®^ International LP (Plymouth Meeting, Pennsylvania, USA. 2′,7′-Dichlorodihydrofluorescein diacetate (DCFH-DA) and high-quality dihydrorhodamine 123 (DHR 123) were purchased from Molecular Probes (Eugene, Oregon, USA) and ONOO^−^ was purchased from Cayman (Ann Arbor, Michigan, USA). The fluorescent D-glucose analogue and glucose tracer 2-[N-(7-nitrobenz-2-oxa-1, 3-diazol-4-yl) amino]-2-deoxy-D-glucose (2-NBDG) was purchased from Life Technologies (Carlsbad, California, USA). Dithiothreitol (DTT) was purchased from Bio-Rad Laboratories (Hercules, California, USA). Fetal bovine serum (FBS), penicillin-streptomycin, minimum essential medium (MEM), 0.25% trypsin-ethylenediaminetetraacetic acid (EDTA), nonessential amino acids, and sodium pyruvate were purchased from Gibco-BRL Life Technologies (Grand Island, NY, USA). β-Actin, secondary antibodies, and PTP1B were obtained from Santa Cruz Biotechnology (Dallas, TX, USA). All other solvents and chemicals were purchased from E. Merck, Fluka, or Sigma-Aldrich, unless otherwise stated.

### 2.3. Plant Materials

Entire plants of *A. decursiva* was purchased from the Korean plant extract bank, which is affiliated with the Korea bioscience and biotechnology research institute, Korea. The whole plant has been documented and deposited for reference in the Department of Food Science and Nutrition, Pukyong National University, Busan, South Korea (Professor Choi, J.S.).

### 2.4. Extraction, Fractionation, and Isolation

All plants powder of *A. decursiva* (2.9 kg) was refluxed for 3 h (3 × 10 L) with methanol (MeOH). In order to produce the MeOH extract (763.32 g), the total filtrate was then concentrated until dried in vacuo at 40 °C. This extract was suspended in distilled water and subsequently divided into dichloromethane (CH_2_Cl_2_), ethyl acetate (EtOAc), and *n*-butanol (*n*-BuOH) for the processing of CH_2_Cl_2_ (210.47 g), EtOAc (15.06 g) and *n*-BuOH (159.07 g) fractions, respectively, in addition to the water residue (444.32 g). Initially, the CH_2_Cl_2_ fraction (210.47 g) was subjected to silica gel column chromatography and eluted with CH_2_Cl_2_, followed by the addition of gradually increasing amounts of MeOH, to create 20 subfractions (CF01–CF20). Fraction 4 (CF04) has also been chromatographed over silica gel using hexane: EtOAc (10:1) in order to obtain three subfractions (CF05F1–CF05F3). Fraction 1 of fraction 5 (CF05F01) was chromatographed using CH_2_Cl_2_ over silica gel to obtain Pd-C-I (200 mg) and Pd-C-III (92 mg). Fraction 2 of fraction 5 (CF05F03) was also chromatographed over silica gel using CH_2_Cl_2_ to obtain Pd-C-II (42 mg). Fraction 6 (CF06, 2.33 g) was chromatographed with *n*-hexane:EtOAc (50:1 to10:1, gradient) on a silica gel column to obtain (+)-*trans*-decursidinol (130.13 mg). All compounds were previously isolated and identified in our laboratory [33,34,36,45].

### 2.5. Assay for PTP1B Inhibitory Assay

The inhibitory activity of PTP1B has been tested using *p*NPP previously reported in our paper [31]. The amount of *p*-nitrophenyl produced after enzymatic dephosphorylation of pNPP was calculated using a microplate spectrophotometer (Molecular Devices) to calculate absorbance at 405 nm. As a positive control, ursolic acid, was used.

### 2.6. α-Glucosidase Inhibitory Assay

The enzyme inhibition analysis was carried out using the technique previously reported in our paper [31]. The absorbance was recorded at 405 nm immediately afterward using the microplate spectrophotometer (Molecular Devices, Sunnyvale, CA, USA). As a positive control, acarbose dissolved in 10% DMSO was used.

### 2.7. Inhibition of ONOO^−^-Mediated Protein Tyrosine Nitration

ONOO^−^-mediated protein tyrosine nitration was evaluated with minor modifications by using in our previously reported method [31].

### 2.8. Assay for ONOO^−^ Scavenging Activity

ONOO^−^ scavenging activity was measured using a slight modification of the method reported by Kooy’s [46] that monitors output of fluorescent rhodamine 123 from non-fluorescent DHR 123 in the presence of ONOO^−^. The rhodamine buffer (pH 7.4) consisted of 90 mM sodium chloride, 50 mM sodium phosphate dibasic, 5 mM potassium chloride, 100 µM diethylenetriaminepentaacetic acid and 50 mM sodium phosphate monobasic. DHR 123 final concentration was 5 µM. The buffer used in this assay was prepared and put on ice before use. In 10% DMSO, the samples were dissolved and authentic ONOO^−^ (10 µM) dissolved in 0.3 N sodium hydroxide was added. Background and final fluorescent intensities were measured after 5 min. The fluorescence intensity of the oxidized DHR 123 was measured with a microplate fluorescence reader (Bio-Tek Instruments, Inc., FLx 800, Winooski, UT, USA) on the excitation and emission wavelengths of 480 and 530 nm, respectively. As a standard control, l-penicillamine was used.

### 2.9. Total ROS Generation Inhibitory Activity Assay

The generation of ROS was calculated using the ROS-sensitive fluorescence indicator (DCFH-DA) of 7-dichlorofluorescein diacetate [47]. In a 50 mM potassium phosphate buffer, rat kidney homogenates (Tissue bank from Pusan National University, Busan, S. Korea) were prepared from the kidneys of newly killed male Wistar rats weighing between 150–200 g and 10 μg of each test sample was applied to 190 μL of kidney post mitochondrial fraction. The mixture was loaded in a potassium phosphate buffer with 50 μL of DCFH-DA (12.5 mM) and shaken for 5 min. On a microplate fluorescence spectrophotometer FLX 800 (Bio-Tek Instruments Inc., Winooski, UT, USA), the fluorescence of 2′,7′-dichlorofluorescein (DCF), the oxidation product from DCFH-DA, was measured at an excitation wavelength of 485 nm and an emission wavelength of 530 nm for 30 min. As a positive control, Trolox, was used.

### 2.10. Measurement of Intracellular Reactive Oxygen Species Level

A fluorometric assay was used to measure the intracellular ROS level in HepG2 cells applying the oxidant-sensitive fluorescence probe 2, DCFH-DA [47]. In order to determine the extent of intracellular ROS scavenging activity, HepG2 cells were seeded into black 96-well plates at a density of 2.0 × 10^5^ cells/well and incubated with coumarins at a concentrations range of 2.5–40 µM for 1 h. To induce ROS production, cells were subsequently exposed to 200 µM *tert*-butyl hydroperoxide (*t*-BHP) for 30 min and were then incubated for 30 min with DCFH-DA (20 μM). At an excitation wavelength of 485 nm and an emission wavelength of 530 nm, the resulting fluorescence intensities were measured using a fluorescence microplate reader (FL × 800; Bio-Tek Instruments Inc., Winooski, UT, USA).

### 2.11. Kinetic Parameters of Coumarins in Lineweaver-Burk and Dixon Plots for Both PTP1B and α-Glucosidase Inhibition

Two kinetic methods using Lineweaver-Burk and Dixon plots have been complemented for the determination of the kinetic process [48,49,50]. PTP1B inhibition mode was calculated at different pNPP substrate concentration (0.5, 1.0, and 2.0 mM) in the absence or presence of different coumarins test concentrations (0, 0.8, 4.0, and 20.0 µM for (+)-*trans*-decursidinol; 0, 2.0, 10.0, and 50 µM for Pd-C-I and Pd-C-II; 0, 4.0, 20.0, and 100 µM for Pd-C-III) using Lineweaver-Burk and Dixon plots. By monitoring the effects of different concentrations of the substrates in the Dixon plots (single reciprocal plot), each enzymatic inhibition of test coumarins was assessed. Additionally, in the presence of different concentrations of substrate: 2.5, 1.25, and 0.625 mM *p*NPG, the Lineweaver-Burk and Dixon plots for α-glucosidase inhibition of coumarins were also performed. The test concentrations of the coumarins in the α-glucosidase kinetic analysis were as follows 0, 7.81, 15.62, 31.25, and 62.5 µM for (+)-*trans*-decursidinol, Pd-C-I, and Pd-C-II; 0, 15.62, 31.25, 62.5, and 125 µM for Pd-C-III. The PTP1B and α-glucosidase assay methods were the same for both enzymatic procedures. The inhibition constants (K*i*) were calculated by interpretation of the Dixon plots, where the value of the x-axis means -K*i*.

### 2.12. Molecular Docking Simulation of PTP1B and α-Glucosidase Inhibition

The structure of the PTP1B was obtained from the RCSB Protein Data Bank website with its selective catalytic inhibitor 3-({5-[(*N*-acetyl-3-{4-[(carboxycarbonyl)(2-carboxyphenyl)amino]-1-naphthyl}-l-alanyl)amino]pentyl}oxy)-2-naphthoic acid (compound **23**) [51]. By comparison, the α-glucosidase structure was obtained from the website of RCSB Protein Data Bank with its catalytic ligand of acarbose [52]. Accelrys Discovery Studio 16.1 (Accelrys, Inc., San Diego, CA, USA) was used for the preparation of protein. The binding areas of the compound **23** and acarbose were considered the most convenient ligand binding regions for the docking simulation. The 3D structures for (+)-*trans*-decursidinol, Pd-C-I, Pd-C-II, and Pd-C-III were derived from PubChem Compound (NCBI) and with MarvinSketch (Chem Axon, Budapest, Hungary). A Lamarckian genetic algorithm (GA) protocol has been used for docking. By default, gasteiger charges have been added, the rotatable bonds have been set with ADT, and all torsions will rotate. The scale of the grid box size was set to the limit by default spacing. For PTP1B and 21.272, −0.751, 18.633 for α-glucosidase, the X, Y, Z center was 37.303, 30.97, 33.501. The docking simulation was performed with the default parameters 10 independent GA. The results were analyzed using UCSF Chimera (http://www.cgl.ucsf.edu/chimera/) in LigPlot^+^ for the visualization of the interacting residue and the residual interaction with van der Waals and hydrogen bonds.

### 2.13. Assay for Cell Viability 

The HepG2 (human hepatocarcinoma) cell line was purchased from American Type Culture Collection (HB-8065, Manassas, VA, USA). As mentioned earlier [6], cell viability has been assessed using the MTT assay. In short, at a density of 2 × 10^5^ cells/well, HepG2 cells were seeded into a 96-well plate and incubated for 24 h at 37 °C. The cells were fed fresh serum-free MEM, which were incubated for 24 h, with a different concentration of coumarins (5–40 μM). The cells were then incubated in phosphate-buffered saline (PBS) with 100 μL of MTT solution at 0.5 mg/mL and incubated for 2 h. The medium was replaced with 100 μL DMSO (100%) to calculate the proportion of surviving cells. A microplate reader (Molecular Devices, Sunnyvale, CA, USA) measured the resulting absorbance values at 570 nm.

### 2.14. Induction of Insulin Resistant HepG2 Cells

With minor modifications, the insulin resistant HepG2 cell model was developed according to the previous method [6]. In short, in 96-well plats (2 × 10^5^ cells/well), HepG2 cells were cultured. When they reached a confluence, 10^−6^ M insulin was treated for 24 h on serum-free MEM to induce insulin resistance, then the cells were incubated for 30 min with 100 nM insulin and harvested for assays as described below.

### 2.15. 2-NBDG Glucose Uptake Assay

In accordance with a previous method [6] the glucose uptake assay was established with some modifications. Briefly, HepG2 cells were cultured in 96-well plate to confluence and then treated with 10^−6^ M insulin for 24 h to induce insulin resistance (based on a 49% decrease in glucose uptake) (data not shown). Different coumarins (5–40 µM) or 10 µM metformin (non-cytotoxic concentration) concentrations were added and incubated for 24 h, followed by 100 nM insulin and cells incubation for 30 min, respectively. 2-NBDG uptake was measured following this incubation. The cells were incubated with 40 μM of 2-NBDG (dissolved in PBS with 1% BSA) for 20 min. The cells were washed thrice with ice-cold PBS to avoid the reaction, and the fluorescence intensity of 2-NBDG was calculated at 485 nm of excitation and 528 nm of emission on a microplate reader (Bio-Tek Instruments Inc., Winooski, VT, USA). Five replication wells have been set up, and three times each experiment has been replicated.

### 2.16. Cell Lysates Preparation and Analysis of Western Blot

Insulin-resistant HepG2 cells (5 × 10^5^ cells/well) were treated for 24 h in 6-well plates with different concentrations of coumarins. The cells were washed with ice-cold PBS three times, collected, then lysed with a sample buffer [50 mM (4-(2-hydroxyethyl)-1-piperazineethanessulfonic acid) pH 7.5, 150 mM NaCl, 2.5 mM EDTA, 0.5% NP-40, 1 mM PMSF, 1 mM DTT, 0.2% aprotinin, 0.5% leupeptin, 20 mM NaF, and 1 mM Na_3_VO_4_] on ice after stimulation with 100 nM insulin at 37 °C for 30 min. Insoluble materials were extracted by centrifugation for 20 min at 25,000 × *g* after incubation for 30 min. An updated Bradford protein assay kit calculated the protein concentrations. Total protein (40 μg) was electrophoresed into a polyvinylidene difluoride (PVDF) membrane by use of a dodecyl sulfate-polyacrylamide gel electrophoresis (SDS-PAGE). In a blocking buffer, membranes were blocked (5% skim milk in Tris-buffered saline containing 0.1% Tween 20 (TBST) and incubated overnight at 4 °C with primary antibodies, followed by incubation at room temperature for 4 h with suitable secondary antibodies. Three times for 30 min with TBST, the membranes were washed. The Supersignal West Pico chemiluminescence substrate (Pierce, Rockford, IL, USA) has been used to visualize the protein bands and pictured on X-ray film (Kodak, Rochester, NY, USA) according to the manufacturer’s instructions. Using CS analyzer software (Atto Corp., Tokyo, Japan), bands were scanned and quantitated and the control was set to 1. As the internal control, β-actin was used. For determining the molecular weights of protein bands, prestained blue protein markers were used. Comparisons were made between mean band densities in same gels. Autoradiography visualized protein bands and measured the intensities using Quantity One software (Bio-Rad Laboratories, Hercules, CA, USA).

### 2.17. Statistical Analysis

The mean ± SEM of triplicate experiments was represented in all the experiments. A variance (ANOVA) and Duncan’s test (Systat Inc., Evanston, IL, USA) test were used to evaluate statistically significant values. A statistically significant *P*-value of <0.05 was considered.

## 3. Results

### 3.1. Coumarins’ Activity in Inhibiting PTP1B and α-glucosidase

The inhibitory ability of PTP1B and α-glucosidase was evaluated using *p*NPP and *p*NPG as a substrate to evaluate the anti-diabetic activity of the four dihydroxanthyletin-type coumarins (Figure 1) extracted from *A. decursiva*, and the results are expressed as IC_50_ values and presented in Table 1. (+)-*Trans*-decursidinol had the highest inhibitory activity of PTP1B among the isolated compounds, with an IC_50_ of 2.33 ± 0.07 µM, which was three-fold higher than the positive control, ursolic acid (IC_50_ = 6.87 ± 0.19 µM). In addition, Pd-C-I, Pd-C-II, and Pd-C-III exhibited potent PTP1B inhibitory activity, with corresponding IC_50_ values of 4.32 ± 0.12, 6.17 ± 0.31, and 11.98 ± 0.43 µM, respectively. Similarly, as a reference compound, the α-glucosidase inhibition assay was validated with acarbose (IC_50_; 169.49 ± 3.25 µM). A 5–15-fold increase in α-glucosidase inhibition compared to acarbose was observed in the coumarins studied. Among them, (+)-*trans*-decursidinol had a promising inhibitory effect of α-glucosidase with an IC_50_ value of 11.32 ± 0.56 µM, followed by Pd-C-I (IC_50_; 17.40 ± 0.33 µM), Pd-C-II (IC_50_; 24.74 ± 0.89 µM), and Pd-C-III (IC_50_; 36.77 ± 1.04 µM). To determine the structure-activity relation between the dihydroxanthyletin-type coumarins and the target enzyme inhibition, their inhibitory effects on PTP1B and α-glucosidase were investigated. Therefore, given the coumarins’ PTP1B and α-glucosidase inhibitory activities, it can be speculated that the presence of a hydroxyl group at the 4′ position on the chroman ring plays an important role in the dihydroxanthyletin-type coumarins’ activity, and that substituting the hydroxyl group with other functional groups (angeloyl, acetyl or senecioyl) decreased activity. The coumarins (+)-*trans*-decursidinol and Pd-C-I were more active than were the others, implying that the free hydroxyl group at the 4′ position has more influence on PTP1B and α-glucosidase inhibition than does the presence of acetyl or senecioyl groups at the 4′ positions of dihydroxanthyletin-type coumarins. Pd-C-II, which has a free hydroxyl group at the 3′ position, showed potent activities that inhibited PTP1B and α-glucosidase. Compared to our previous [31], and current research, coumarin Pd-C-I showed substantial increase in PTP1B and α-glucosidase inhibitory activity in the free hydroxyl group at the 4′ position, and the substitute methoxy group at the same position coumarin 4′-methoxy Pd-C-I, significantly reduce the activity. In addition, in the presence of an extra hydroxyl group at the 4′ position, coumarin (+)-*trans*-decursidinol was 25-times more active than decursinol. On the other hand, coumarins Pd-C-III and decursidin are structurally similar, but only difference between senecioyl groups present in decursidin and angeloyl and acetyl groups present in Pd-C-III is that the senecioyl group plays a significant role in the inhibition of PTP1B. Based on the structure-activity relation in our previous [31], and current research indicated that the PTP1B and α-glucosidase inhibitory activities appear to depend largely on the presence of the hydroxyl group at the C-4′ position, in which methylation/acetylation and senecioylation at this position diminished the inhibitory potency greatly. Thus, the substituent at the C-4′ position is the factor in obtaining improved potency of PTP1B and α-glucosidase inhibitors.

Our structure-activity relationship allows us understand the main structural elements affecting the PTP1B and α-glucosidase inhibitory activity of the dihydroxanthyletin-type coumarins’.

### 3.2. PTP1B and α-Glucosidase Inhibitory Enzyme Kinetics 

A kinetic analysis of both enzymes at different substrate concentrations was performed to determine the mode by which the coumarins inhibit PTP1B and α-glucosidase. The mode of enzyme inhibition determined by Lineweaver-Burk plots is presented in Table 1. As shown Appendix A, the (+)-*trans*-decursidinol, Pd-C-I, and Pd-C-II appeared to be competitive inhibitors for the PTP1B enzyme with K*i* values of 2.84 µM for (+)-*trans*-decursidinol, 6.21 µM for Pd-C-I, and 9.72 µM for Pd-C-II. Double reciprocal plots in the competitive inhibition mode provided a group of lines with the same *y*-intercept (constant *V*_max_), but a different *x*-intercept (varying *K*_m_ values). In addition, Pd-C-III showed a mixed-type inhibition of PTP1B (as the inhibitor concentration increased, *K*_m_ increased and *V*_max_ decreased) with a *K*_i_ value of 16.04 µM. Appendix A, show the enzymatic kinetic results of (+)-*trans*-decursidinol, Pd-C-I, Pd-C-II, and Pd-C-III in α-glucosidase inhibition. (+)-*trans*-decursidinol exhibited competitive-type α-glucosidase inhibition, while Pd-C-I and Pd-C-II displayed mixed-type inhibition, with *K*_i_ values of 16.02, 22.98, and 30.83 µM, respectively (Table 1), and Pd-C-III showed noncompetitive-type α-glucosidase inhibition with a K*i* value of 55.59 µM.

### 3.3. PTP1B Inhibition Molecular Docking Simulation

A molecular docking simulation of AutoDock 4.2 was used to investigate the binding position of inhibitors within the active site of PTP1B (Appendix A). A molecular docking simulation of (+)-*trans*-decursidinol, Pd-C-I, Pd-C-II, and Pd-C-III with PTP1B was conducted in these experiments, in which 3-({5-[(N-acetyl-3-{4-[(carboxycarbonyl)(2-carboxyphenyl)amino]-1-naphthyl}-L-alanyl)amino] pentyl}oxy)-2-naphthoic acid (compound **23**) was considered the standard ligand for validating the results of the simulation. The binding energies of dihydroxanthyletin-type coumarins with interacting residues, including H-bond and van derWaals residues, together with the number of H-bonds, are listed in Table 2. For PTP1B, the modelling results suggested that (+)-*trans*-decursidinol formed five H-bonds with Ala27, Ile219, Gly220, Arg221, and Gly183, with a binding energy of –7.9 kcal/mol. In addition, hydrophobic interactions with Cys215, Gln266, Phe182, Thr263, and Ser216 residues stabilized the PTP1B-(+)-*trans*-decursidinol complex (Figure 2A). 

The PTP1B-Pd-C-I inhibitor complex showed, as shown in Figure 2B, a –7.8 kcal/mol binding energy with six H-bonds and Gln266, Arg221, and Trp179 interacting residues. As illustrated in Figure 7B, Pd-C-I’s corresponding ligand interactions at PTP1B’s active site were the six H-bonding interactions between the Arg221 and Trp179 enzyme residues and the senecioyl group and Gln266 residue with Pd-C-I’s hydroxyl group. In addition, hydrophobic interactions were also observed between the Asp48, Gln262, Gly163, Ile219, Lys116, Thr263, Tyr46, and Val49 residues. Moreover, Pd-C-II’s binding affinity for PTP1B was –7.5 kcal/mol for six H-bonds with interacting Cys215, Ala217, Ile219, Gly183, and Arg221 residues (Figure 2C). The Pd-C-II binding involved the formation of three H-bonds between the Cys215, Ala217, and Ile219 residues of PTP1B and the interacting Pd-C-II ketone group and Gly183 and Arg221 residues with a hydroxyl group and oxygen atom, respectively. Furthermore, hydrophobic associations were formed by the residues Asp181, Gln266, Gly220, Gln262, Pro180, Trp179, and Thr263, thus enhancing the protein-ligand interaction between PTP1B and Pd-C-II. Pd-C-III exhibited a –7.7 kcal/mol binding affinity for the allosteric site of PTP1B, as illustrated in Figure 2D. In addition, three H-bonds interactions with the Ile219, Ala217, and Cys215 residues and Arg221 residue with a Pd-C-III oxygen atom were shown by the interacting ketone group. Hydrophobic interactions were observed between the Pd-C-III and PTP1B residues Asp181, Gly183, Gln266, Gln262, Gly220, Lys116, Phe182, Pro180, and Trp179, which stabilized the protein-ligand interaction further.

### 3.4. Molecular Docking Simulation of α-Glucosidase Inhibition

A molecular docking simulation of (+)-*trans*-decursidinol, Pd-C-I, Pd-C-II, and Pd-C-III with α-glucosidase was performed with AutoDock 4.2, and the ligand-enzyme complexes of the four coumarins/or acarbose were posed stably in the same pocket of the α-glucosidase (PDB ID: 3A4A) (Appendix A). Table 3 provides a description of the binding energies of the coumarins and reference ligands studied, along with a list of the amino acid residues involved in the H-bonds and hydrophobic interactions. The α-glucosidase-(+)-*trans*-decursidinol inhibitor complex displayed a binding energy of −7.09 kcal/mol according to the AutoDock 4.2 simulation results shown in Figure 3A, and contained three hydrogen bonds with the interacting residues Trp229, Ans496, and Asp232. In addition, hydrophobic interactions with Ile503, Thr507, Lys506, Ala231, Ile233, Ser497, Asn230, and Ser505 stabilized the α-glucosidase-(+)-*trans*-decursidinol complex. Moreover, the Pd-C-I binding affinity for α-glucosidase was −8.09 kcal/mol for four H-bonds with interacting Trp229, Asp232, Ans496, and Lys506 residues (Figure 3B). Hydrophobic interactions were also observed between Pd-C-I and the Asn230, Ala231, Thr507, Ile503, Ser505, Asn475, Ile233, and Phe476 residues. On the other hand, the inhibitor complex of α-glucosidase-Pd-C-II had −7.6 kcal/mol binding energy for four hydrogen bonds with the interactive residues of Thr409, Arg412, and Gly408 at the allosteric site of α-glucosidase. In addition, hydrophobic interactions were also observed between the Ile460, Leu369, Thr368, Lys365, Tyr407, Ser406, Phe457, Glu453, Arg456, Phe367, and Leu376 residues (Figure 3C). Pd-C-III’s binding affinity for α-glucosidase was −7.24 kcal/mol for three hydrogen bonds with interacting Asp232, Ser497, and Ser505 residues. In addition, the residues of Ile503, Thr507, Ala231, Lys506, Ile233, Asn237, Val501, Asn496, His524, Asn230, and Trp229 were involved in certain hydrophobic interactions (Figure 3D). As shown in Figure 3E, and Table 3, the docking results showed that acarbose (a catalytic inhibitor) formed seventeen hydrogen bonds with the Ser240, Gln182, Glu277, Asp352, Arg442, Asp307, His280, Asp69, His112, Asp242, Tyr158, Arg213, and Asp215 residues.

### 3.5. Coumarins’ Effect on Glucose Uptake in HepG2 Insulin-Resistant Cells

In order to investigate the capacity of coumarins’ for increasing glucose uptake, the MTT assay tested the cytotoxicity of four coumarins’ to HepG2 cells. A 2-[N-(7-nitrobenz-2-oxa-1,3-diazol-4-yl) amino]-2-deoxyglucose (2-NBDG) test was used for testing of glucose uptake in insulin-resistant HepG2 cells. Treatment of high-dose of insulin-induced insulin resistance HepG2 cells as demonstrated by a substantial reduction in glucose uptake in the insulin-resistant group (Figure 4). However, all test compounds significantly increased dose-dependent uptake of insulin-stimulated 2-NBDG in insulin-resistant HepG2 cells. Among them, the most powerful effect was (+)-*trans*-decursidinol, followed by Pd-C-I, Pd-C-II, and Pd-C-III. The relative glucose uptake percentages of a 15 µM concentration of (+)-*trans*-decursidinol was 125%, and 40 µM concentrations of Pd-C-I, Pd-C-II, and Pd-C-III were 128%, 120%, and 117%, respectively, which were comparable to the positive control, metformin (relative % glucose uptake = 129% at 10 µM). Taken together, these results suggested that dihydroxanthyletin-type coumarins are able to induce insulin-resistant HepG2 cells’ sensitivity to insulin and increase the relative percentage of glucose uptake thereby.

### 3.6. Effects on PTP1B Expression Level in Insulin-Resistant HepG2 Cells

Insulin signaling is negatively regulated by PTP1B and increased activity and expression are implicated in the insulin resistance pathogenesis. We have assessed its expression in insulin-resistant HepG2 cells by Western blot, to confirm that these coumarins improve insulin sensitivity by inhibiting expression of PTP1B. All dihydroxanthyletin-type coumarins inhibited PTP1B expression in a dose-dependent manner, which disclosed their insulin-sensitizing potential, as shown in Figure 5. (+)-*trans*-decursidinol diminished PTP1B’s expression at 10 and 15 µM, while Pd-C-I and Pd-C-II downregulated PTP1B’s expression at 20 and 40 µM. The order of the PTP1B inhibitory expression level was (+)-*trans*-decursidinol > Pd-C-I > Pd-C-II > Pd-C-III.

### 3.7. The Coumarins’ Inhibitory Effect on ONOO^−^-Mediated Albumin Nitration

Western blot analysis was carried out using 3-nitrotyrosine antibody to determine the inhibitory effect of these dihydroxanthyletin-type coumarins’ on albumin’s ONOO^—^-induced nitration. As shown in Figure 6, ONOO^—^-mediated albumin nitration was significantly inhibited in a concentration-dependent manner by pretreatment with (+)-*trans*-decursidinol at various concentrations (5–40 µM). In addition, pretreatment at different concentrations (12.5–100 µM) with Pd-C-I, Pd-C-II, and Pd-C-III resulted in a pronounced dose-dependent inhibition of ONOO^—^ -mediated albumin nitration (Figure 6).

### 3.8. The Effect of Coumarins’ on the Levels of Intracellular ROS in t-BHP-Treated HepG2 Cells

Next, we investigated their effects on ROS production in HepG2 cells exposed to *t*-BHP (200 μM) in order to determine if the anti-diabetic effects of dihydroxanthyletin-type coumarins’ could be due to a reduction oxidative stress. Exposure to *t*-BHP (200 μM) modifies the cells redox state, resulting in the formation of ROS. We observed that when the cells were treated with *t*-BHP (200 μM), ROS generation increased significantly to levels of approximately 100%, suggesting that *t*-BHP had a powerful effect on ROS generation in HepG2 cells. ROS production decreased in a co ncentration-dependent manner when the HepG2 cells were pretreated with (+)-*trans*-decursidinol, with 75.53%, 57.55%, and 43.44% inhibition at concentrations of 5, 10, and 15 μM, respectively (Figure 7A). In addition, Pd-C-I, Pd-C-II, and Pd-C-III significantly inhibited ROS generation at concentrations of 10–40 μM and protected cells from ROS-induced oxidative stress in a concentration-dependent manner compared to *t*-BHP-induced HepG2 cells, as shown in Figure 7B–D. The positive control, Trolox, inhibited ROS production by about 49% at 10 μM. Such findings have clearly shown that dihydroxanthyletin-type coumarins serve as ROS generation scavengers induced by *t*-BHP in HepG2 cells.

### 3.9. The Coumarins’ Inhibitory Effect on ROS and ONOO^−^

To evaluate the coumarins’ antioxidant effect, their potential to inhibit ROS and ONOO− was evaluated (Table 4). The coumarins (+)-*trans*-decursidinol, Pd-C-I, Pd-C-II, and Pd-C-III, showed promising ONOO^−^ scavenging activity, with IC_50_ values of 4.57 ± 0.21, 1.82 ± 0.44, 3.77 ± 0.15, and 1.44 ± 0.22 μM, respectively, while the positive control, *l*-penicillamine, had an IC_50_ value of 1.86 ± 0.11 µM. In addition, (+)-*trans*-decursidinol, Pd-C-I, Pd-C-II, and Pd-C-III exhibited strong ROS scavenging effects, with IC_50_ values of 9.77 ± 0.32, 6.32 ± 0.41, 10.76 ± 0.73, and 9.69 ± 0.21 µM, respectively, while the positive control, Trolox, had an IC_50_ value of 3.29 ± 0.26 µM.

## 4. Discussion

Diabetes mellitus (DM) is a chronic metabolic disease that has a global effect on the health of humans. Hyperglycemia is the hallmark of diabetes attributable to either insulin resistance or insulin deficiency, or both [53]. Several hypoglycemic agents with different mechanisms of action have been developed, such as decreasing the production of hepatic glucose, increasing insulin production, inhibiting gluconeogenesis, and reducing postprandial glucose absorption. However, these agents had a range of limitations/side effects, such as gastrointestinal complaints, weight gain, headache, peripheral edema, and hypotension [54]. In addition, the therapies commonly used are minimal, expensive, and some are, actually, unproductive. New approaches to finding more active drugs with various modes of action to treat this life-threating illness are therefore urgently required. 

The most important approach for T2DM treatment is optimum blood glucose levels after a meal. A more active approach to T2DM treatment has recently been involved in the disruption of dietary monosaccharide absorption by inhibiting α-glucosidase [55] and PTP1B, which is involved in the dephosphorylation and inactivation of insulin receptors and attenuates insulin signaling [56]. PTP1B plays a crucial function in phosphorylating and inactivating the insulin receptor to shut down insulin signaling. Insulin receptors and insulin-receptor substrate-1 interact directly to hydrolyze phosphorylated tyrosine, leading in impaired glucose uptake [57]. Furthermore, reducing PTP1B expression and activity was enough to boost the insulin signaling pathway and improve insulin sensitivity [10,11]. A PTP1B inhibitor is therefore expected demonstrate anti-diabetic effects by increasing insulin sensitivity in T2DM [57].

Another approach to treating diabetes is to delay the absorption of glucose by inhibiting an α-glucosidase carbohydrate-hydrolyzing enzyme, which is the main enzyme to catalysing the end-stage metabolism of carbohydrate, especially for non-insulin-dependent diabetes (postprandial hyperglycemia). The inhibitory action of this enzyme slows the production of D-glucose from dietary carbohydrates, delays the glucose uptake/reduces glucose levels and ultimately eliminates postprandial hyperglycemia. However, gastrointestinal side effects are associated with exiting clinical α-glucosidase inhibitors (acarbose, miglitol, and voglibose) [58]. New phytochemicals with strong inhibitions of PTP1B and α-glucosidase with no or less side effects are therefore necessary urgently.

*A. decursiva* L. is a well-known TCM herb of medicinal value for decades, and for medicinal purposes, the whole plant, leaves, and root parts of this plant were used. In our ongoing research into new diabetes phytochemicals, we have assessed the potential of the four major dihydroxanthyletin-type coumarins which are isolated from the whole plant [33,34] to inhibit PTP1B and α-glucosidase. All of the compounds displayed good dual inhibition of both enzymes and were superior to the positive control, while (+)-*trans*-decursidinol, Pd-C-I, Pd-C-II, and Pd-C-III were potent inhibitors of PTP1B. *A. decursiva* is a rich source in different types of coumarin derivatives, including dihydroxanthyletin, furanocoumarin, psoralen, pyranocoumarin, dihydropsoralen, hydroxycoumarin, and dihydropyran [35]. Among the different classes of compounds, dihydroxanthyletin-type coumarins showed promising activity against PTP1B and α-glucosidase compared to previous and current studies. Moreover, we also observed the structure-activity relationship between the dihydroxanthyletin-type coumarins for PTP1B and α-glucosidase inhibition. Therefore, taking into account the inhibitory activity of dihydroxanthyletin-type coumarins with PTP1B and α-glucosidase, it can be speculated that the presence of a hydroxyl group at the C-4′ position has a greater effect on the inhibitory activity of PTP1B and α-glucosidase than the presence of senecioyl/angeloyl/acetyl or the methoxy groups at the C-4′ position of dihydroxanthyletin-type coumarins. Further, we also investigated the inhibition mechanism using an enzyme kinetic study with varying substrate and inhibitor concentrations. These studies revealed that (+)-*trans*-decursidinol, Pd-C-I, and Pd-C-II exhibited competitive-type PTP1B inhibition, while Pd-C-III showed mixed-type inhibition.

Small molecule PTP1B inhibitors are difficult to detect since the PTP1B catalytic pocket is relatively shallow, minimizing the benefit of a small molecule inhibitor; however, the catalytic site consists of many polar amino acid residues (e.g., Ser216, Cys215, and Arg221) that selectively bind small molecules to single groups [59,60]. As we have shown, AutoDock 4.2 stabilized ligand–enzyme complexes in the same PTP1B pocket with four coumarins/or compound **23**. In the corresponding ligand interactions of (+)-*trans*-decursidinol at PTP1B’s active site, there were five hydrogen-bonding interactions with five important residues (Gly183 OH, Arg221 OH, Gly220 CO, Ile219 CO, and Ala217 CO) of the enzyme, and (+)-*trans*-decursidinol also bound at PTP1B’s catalytic site. Three major enzyme residues (Gln266 OH, Arg221 -O-, and Trp179 O atom) participated in six hydrogen-bonding interactions in the case of Pd-C-I. In addition, five important enzyme residues (Cys215 CO, Ala217 CO, Ile219 CO, Gly183 OH, and Arg221 O atom) were involved in H-bonding interactions and residues, in which PTP1B’s catalytic site interacted with Pd-C-II. In addition, between the four coumarins, H-bonds were formed, and the main residues of Arg221 and Cys215 stabilized these coumarins and allowed them to fit together at the catalytic site of the enzyme. Shen et al. [60] also stated that the residues of Arg221 and Cys215 in the formation of hydrogen bonds are essential for catalytic PTP1B inhibitors. Our findings were consistent with those of Barford et al. [59], who indicated the minimum prerequisite for any compounds to be considered a catalytic PTP1B inhibitor is the formation of hydrogen bonds with Arg221. In addition, the hydrogen-bonding interaction was attended by four essential enzyme residues (Ile219, Cys215, Arg221, and Ala217), whereas the allosteric site residues of PTP1B’s site interacting with Pd-C-III. Furthermore, the binding energy of the coumarins’ was negative, implying that hydrogen bonding would stabilize the open form of the enzyme and potentiate tight binding with the active site of PTP1B, resulting in more successful inhibition of PTP1B. Taken together, the findings of kinetic analysis and molecular docking simulation showed that the anti-diabetic properties of these four coumarins were promising.

In addition, with AutoDock 4.2, a molecular docking analysis was performed to explore the binding mode of the coumarins’ inside the α-glucosidase binding pocket to understand their molecular interactions. For α-glucosidase, the docking results suggested that (+)-*trans*-decursidinol’s 4′-OH group formed a strong hydrogen bond with the Asp232 residue’s active site, while a ketone group and oxygen atom also exhibited two important hydrogen bonds with Trp229 and Ans496, and the Pd-C-I and Pd-C-II’s hydroxyl groups exhibited hydrogen interactions with the Asp232 and Gly408 residues. In addition, H-bonds between the four coumarins were formed and the main residues Trp229, Ans496, Lys506, Thr409, Arg412, Asp232, Ser497, and Ser505 stabilized these coumarins at the active site and allowed them to fit into the active pocket of the enzyme. Further, several hydrophobic interactions between the coumarins and active site residues were observed that stabilized their binding at the active site of α-glucosidase. Together, these observations indicate that these four coumarins do not bind to the same α-glucosidase catalytic site as acarbose.

In order to further explore coumarin’s diabetic mechanisms, we have studied the impact of these compounds on the insulin sensitivity in normal versus insulin resistant HepG2 cells. PTP1B is a well-known negative insulin-signalization pathway regulator [10]. An indicator of insulin resistance is increased expression of PTP1B in insulin-sensitive tissues, while lower expression increases, and enhances glucose uptake and insulin signaling [57]. Increased glucose uptake in insulin-resistant HepG2 sample-treated cells compared to the untreated insulin-resistant/normal group has been reported to increase insulin sensitivity. In the 2-NBDG assay, four dihydroxanthyletin-type coumarins significantly increased the dose-dependent absorption of insulin-stimulated glucose. Similarly, (+)-*trans*-decursidinol and Pd-C-I suppressed PTP1B expression in a concentration-dependent manner in Western blot study, while Pd-C-II and Pd-C-III significantly downregulated PTP1B’s expression. In the normal group, PTP1B was expressed weakly, but in the insulin-resistant group, it intensified. While PTP1B inhibition is known to boost insulin sensitivity effectively, there is still a lack of a better understanding of the regulatory mechanism of PTP1B. Some PTP1B inhibitors have been found to inhibit IR and IRS phosphorylation, contribute to insulin resistant, and activates insulin signaling by overexpression of PTP1B [3,6,61,62,63].

ROS and reactive nitrogen species (RNS) are considered to play a central role in DM- associated liver diseases. Therefore, to avoid or alleviate liver diseases associated with oxidative stress, the use of antioxidants has been proposed. It is believed that the antioxidant effects of phenolics, particularly polyphenols, are due at least in part, to the chelation of metal ions, the prevention of radical formation, the indirect modulation of enzyme activity, and modulations of antioxidant and detoxifying enzymes expression [64,65]. Recently, we also reported that polyphenolic compounds (flavonoid) scavenge free radical formation and modulate enzyme activity indirectly [3,6]. All coumarins have considerably scavenged total ROS and have also had a significant effect on the generation of ROS induced by *t*-BHP in HepG2 cells. ONOO^−^ is a highly reactive oxidizing and nitrating agent derived from NO. and ·O_2_- that oxidizes cellular components, such as carbohydrates, proteins, DNA, and lipids [66], and can be indirectly inferred from the presence of nitrotyrosine residues in the formation of ONOO^−^ [22]. Both coumarins have scavenged ONOO^−^ and inhibited ONOO−-mediated tyrosine nitration significantly in a dose-dependent manner. These findings showed that the anti-DM properties of coumarins can be due to the direct hepatocytes scavenging of ONOO^−^ and total ROS.

## 5. Conclusions

In conclusion, the study indicates the ability to treat DM therapeutically with PTP1B and α-glucosidase inhibited by *A. decursiva*-derived dihydroxanthyletin-type coumarins. The evidences showed that coumarins enhanced insulin-sensitivity by downregulating PTP1B and promoted the uptake of glucose in insulin-resistant HepG2 cells. The coumarins also appeared to exert their anti-DM effects by scavenging ONOO^−^ and suppressing ONOO^−^-mediated albumin nitration effectively by reducing total and intracellular ROS production. Therefore, *A. decursiva*-derived dihydroxanthyletin-type coumarins could also supplement pharmacotherapy in the treatment of diabetes. Further, in vivo studies are also important for validating our findings.

## Figures and Tables

**Figure 1 antioxidants-10-00292-f001:**
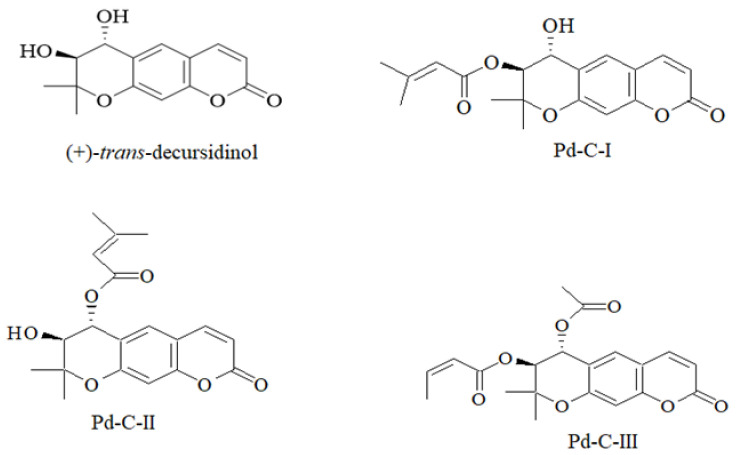
Chemical structure of the isolated dihydroxanthyletin-type coumarins from *Angelica decursiva*.

**Figure 2 antioxidants-10-00292-f002:**
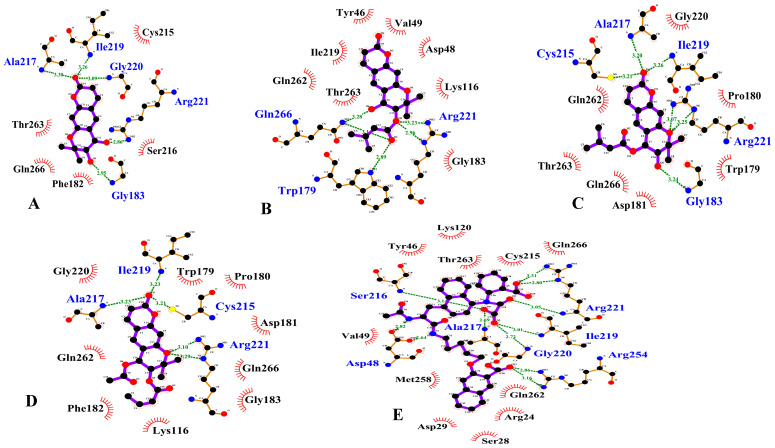
Two dimensional ligand interaction diagram of PTP1B inhibition by (+)-*trans*-decursidinol (**A**), Pd-C-I (**B**), Pd-C-II (**C**), Pd-C-III (**D**), and compound **23** (**E**). Schematic representation of interaction between ligands (coumarins) and PTP1B, coumarins are presented by thick purple stick models, hydrogen bonds are green dotted lines, and hydrophobic interactions with the corresponding amino acid residues of the enzyme are shown by dashed half-moons.

**Figure 3 antioxidants-10-00292-f003:**
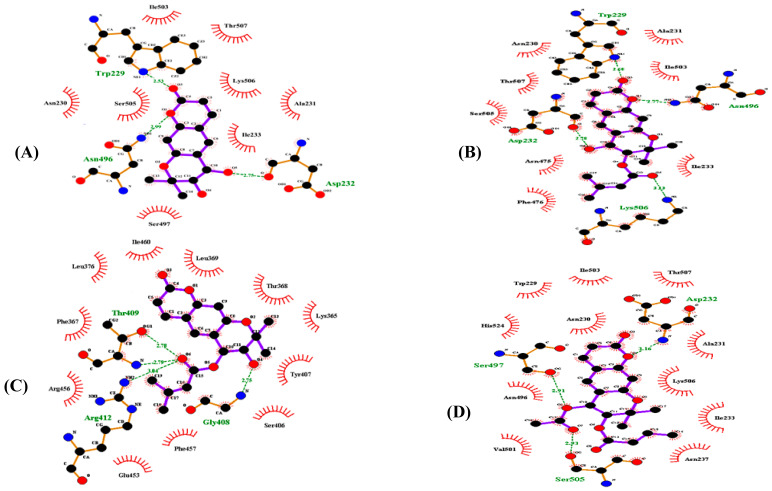
Two-dimensional ligand interaction diagram of α-glucosidase inhibition by (+)-trans-decursidinol (**A**), Pd-C-I (**B**), Pd-C-II (**C**), Pd-C-III (**D**), and acarbose (**E**). Schematic display of the interaction of ligands (coumarins) and α-glucosidase, coumarins present thick purple stick models, hydrogen bonds are green dotted and hydrophobic interactions of dashed half-moons with the enzyme’s respective amino acid residue.

**Figure 4 antioxidants-10-00292-f004:**
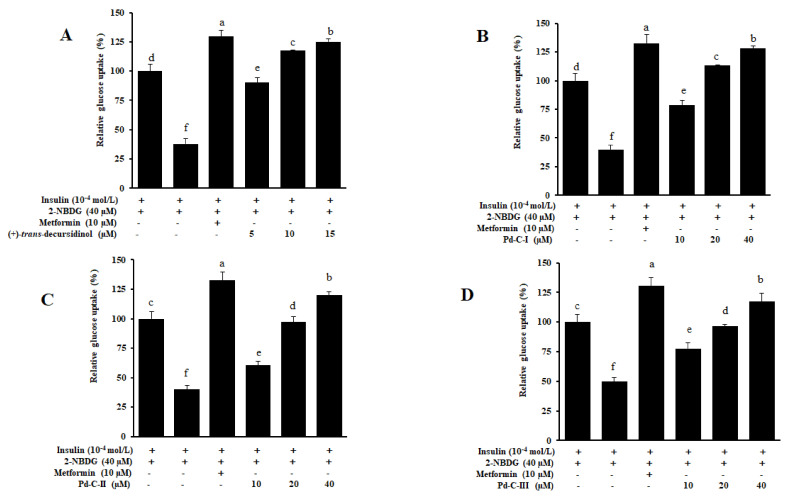
Effect of dihydroxanthyletin-type coumarins on insulin-stimulated glucose uptake in insulin-resistant HepG2 cells (**A**–**D**). The use of fluorescent D-glucose analogue 2-[N-(7-nitrobenz-2-oxa-1, 3-diazol-4-yl) amino]-2-deoxy-D-glucose (2-NBDG) was tested for glucose uptake; insulin at 10^−4^ mol/L was utilized to induce insulin resistance. Different concentrations of dihydroxanthyletin-type coumarins or metformin were treated with cells for 24 h, and insulin-stimulated 2-NBDG uptakes were assessed for 24 h. ^a–f^ Different letters indicate statistical differences among means of each metformin and dihydroxanthyletin-type coumarins concentrations. Results were analyzed by ANOVA and Duncan’s test (*p* < 0.05).

**Figure 5 antioxidants-10-00292-f005:**
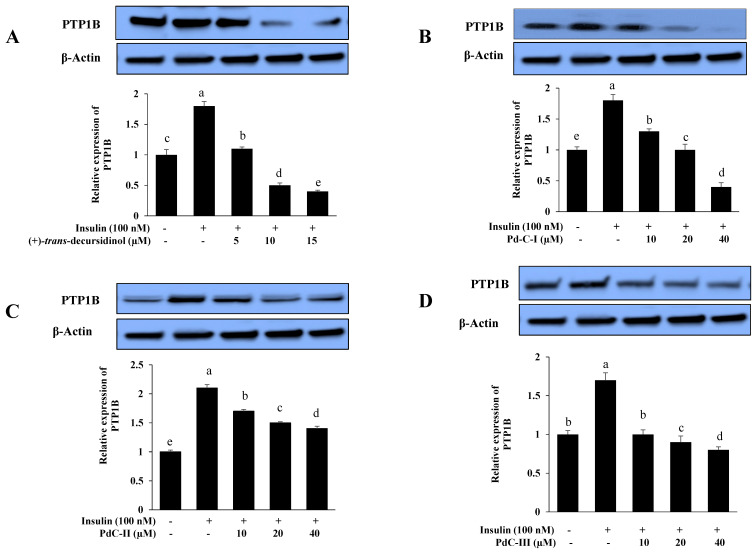
Effect of dihydroxanthyletin-type coumarins (**A**–**D**) on the levels of expression of protein tyrosine phosphatase 1B (PTP1B) in insulin-resistant HepG2 cells. Relative densities of PTP1B versus β-actin. The intensity of protein bands was quantified by densitometry. The results have been normalized versus β-actin levels and are described in three different experiments as the means ± SEMs. ^a–e^ Mean with different letters are significantly differences among means of dihydroxanthyletin-type coumarins concentration. Results were analyzed by ANOVA and Duncan’s test (*p* < 0.05).

**Figure 6 antioxidants-10-00292-f006:**
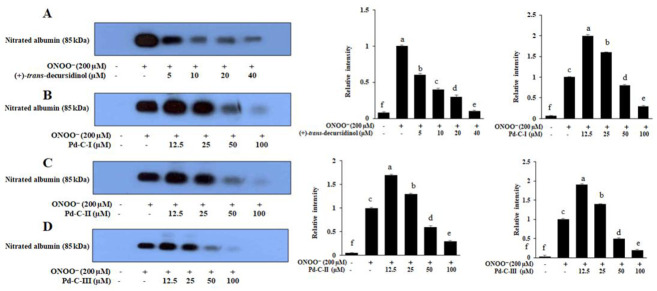
Dose-depended inhibition of ONOO^−^-mediated albumin nitration by dihydroxanthyletin-type coumarins. Test sample mixtures, bovine serum albumin (BSA), and ONOO^−^ were incubated for 30 min with shaking at 37 °C. The reactant was resolved by electrophoresis in a 10% polyacrylamide gel. (**A**) (+)-*trans*-decursidinol; (**B**) Pd-C-I; (**C**) Pd-C-II; and (**D**) Pd-C-III were used at the indicated concentrations. Band intensity quantification has been measured using CS Aanlyzer 3.00 (ATTO Corp, Tokyo, Japan). ^f^ Indicates the untreated normal group, and ^a,c^ indicate significant the ONOO^−^-treated control. ^a–e^ Different letters indicate statistical differences among means of each coumarins concentrations. Results were analyzed by ANOVA and Duncan’s test (*p* < 0.05).

**Figure 7 antioxidants-10-00292-f007:**
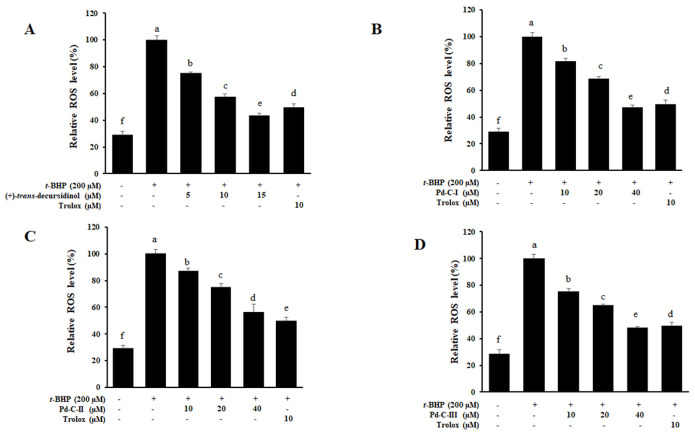
Effects of (+)-*trans*-decursidinol (**A**), Pd-C-I (**B**), Pd-C-II (**C**), and Pd-C-III (**D**) on *tert*-butyl hydroperoxide (*t*-BHP)-induced reactive oxygen species (ROS) generation in HepG2 cells. Dihydroxanthyletin-type coumarins and trolox (10 μM) cells pretreated at var ying concentrations (5 to 40 μM), for 1 h have been stimulated for 30 min with 200 μM *t*-BHP. ROS levels were measured by 2′,7′-dichlorodihydro fluoresce in diacetate with fluorescent analysis. The control values were obtained in the absence of *t*-BHP (200 μM), coumarins after the addition of *t*-BHP (200 μM). As a positive control, trolox was used. Data are expressed as the mean ± SEMs of three independent experiments. ^a–f^ Mean with different letters are significantly different with Duncan’s test at *p* < 0.05. ^f^ Indicates the unstimulated control group, ^a^ Indicates *t*-BHP-treated group.

**Table 1 antioxidants-10-00292-t001:** Protein tyrosine phosphatase 1B (PTP1B) and α-glucosidase inhibitory activities of dihydroxanthyletin-type coumarins from *Angelica decursiva*.

Compounds	Protein Tyrosine Phosphatase 1B			α-Glucosidase		
	IC_50_ (µM) ^a^	*K*_i_ Value ^b^	Inhibition Type ^c^	IC_50_ (µM) ^a^	*K*_i_ Value ^b^	Inhibition Type ^c^
(+)-*trans*-decursidinol	2.33 ± 0.07	2.84	Competitive	11.32 ± 0.56	16.02	Competitive
Pd-C-I	4.32 ± 0.12	6.21	Competitive	17.40 ± 0.33	22.98	Mixed
Pd-C-II	6.17 ± 0.31	9.72	Competitive	24.74 ± 0.89	30.83	Mixed
Pd-C-III	11.98 ± 0.43	16.04	Mixed	36.77 ± 1.04	55.59	Noncompetitive
Ursolic acid ^d^	6.87 ± 0.19	-	-			
Acarbose ^e^				169.49 ± 3.25	-	-

^a^ The 50% inhibition concentration (µM) was measured using a log–dose inhibition curve and is expressed as the mean ± SEM of triplicate studies. ^b^ Determined by the Dixon plot. ^c^ Determined by Dixon and Lineweaver-Burk plots.^d,e^ Used as positive controls.

**Table 2 antioxidants-10-00292-t002:** Molecular interaction of the protein tyrosine phosphatase 1B (PTP1B) active site with dihydroxanthyletin-type coumarins, as well as reference inhibitor.

Compounds	Binding Energies (Kcal/mol) ^a^	No. of H Bond ^b^	H-bonding Interacting Residues ^c^	Van der Waals Interacting Residues ^d^
(+)-*trans*-decursidinol	−7.9	5	Ala27, Ile219, Gly220, Arg221, Gly183	Cys215, Gln266, Phe182, Thr263, Ser216
Pd-C-I	−7.8	6	Gln266, Arg221, Trp179	Asp48, Gln262, Gly163, Ile219, Lys116, Thr263, Tyr46, Val49
Pd-C-II	−7.5	6	Cys215, Ala217, Ile219, Gly183, Arg221	Asp181, Gln266, Gly220, Gln262, Pro180, Trp179, Thr263
Pd-C-III	−7.7	5	Ile219, Ala217, Cys215, Arg221	Asp181, Gly183, Gln266, Gln262, Gly220, Lys116, Phe182, Pro180, Trp179
Compound 23	−11.23	11	Asp48, Ala217, Arg221, Arg254, Gly220, Ile219, Ser216	Arg24, Asp29, Cys215, Gln266, Gln262, Lys120, Met258, Ser28, Thr263, Tyr46, Val49

^a^ The biding free energy of the ligand receptor complex was calculated. ^b,c,d^ The AutoDock 4.2. program was used to evaluate the number of hydrogen bonds and all the amino acid residues of the enzyme inhibitor complex.

**Table 3 antioxidants-10-00292-t003:** Binding sites of dihydroxanthyletin-type coumarins in α-glucosidase using the Autodock 4.2 molecular docking program.

Compounds.	Binding Energies (Kcal/mol) ^a^	No. of H Bond ^b^	H-bonding Interacting Residues ^c^	Van der Waals Interacting Residues ^d^
(+)-*trans*-decursidinol	−7.09	3	Trp229, Ans496, Asp232	Ile503, Thr507, Lys506, Ala231, Ile233, Ser497, Asn230, Ser505
Pd-C-I	−8.09	4	Trp229, Asp232, Ans496, Lys506	Asn230, Ala231, Thr507, Ile503, Ser505, Asn475, Ile233, Phe476
Pd-C-II	−7.6	4	Thr409, Arg412, Gly408	Ile460, Leu369, Thr368, Lys365, Tyr407, Ser406, Phe457, Glu453, Arg456, Phe367, Leu376
Pd-C-III	−7.24	3	Asp232, Ser497, Ser505	Ile503, Thr507, Ala231, Lys506, Ile233, Asn237, Val501, Asn496, His524, Asn230, Trp229
Acarbose	−10.20	17	Arg442, Asp307, Asp215, Arg213, Asp352, Asp69, Asp242, His280, Tyr158, His112, Gln182, Glu277, Ser240,	Arg315, Gln353, His351, Glu411, Tyr72, Lys156, Gln279, Phe178, Phe303, Val216

^a^ Estimated the biding free energy of the ligand receptor complex. ^b,c,d^ AutoDock 4.2. program was used to evaluate the number of hydrogen bonds and all the amino acid residues of the enzyme inhibitor complex.

**Table 4 antioxidants-10-00292-t004:** Inhibitory effects of dihydroxanthyletin-type coumarins on ONOO^−^ and ROS.

Test Sample	IC_50_ (µM) ^a^	
	ONOO^−^	ROS
(+)-*trans*-decursidinol	4.57 ± 0.21	9.77 ± 0.32
Pd-C-I	1.82 ± 0.44	6.32 ± 0.41
Pd-C-II	3.77 ± 0.15	10.76 ± 0.73
Pd-C-III	1.44 ± 0.22	9.69 ± 0.21
l-Penicillamine ^b^	1.86 ± 0.11	-
Trolox ^c^	-	3.29 ± 0.26

^a^ The 50% inhibition concentration (µM) is measured and expressed as the mean ± SEM of triplicate experiments from a log-dose inhibition curve. ^b,c^ Positive controls used in respective assays.

## Data Availability

The data presented in this study are available within the article and its Appendix A.

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
