# Peer review of "Insulin–Mimetic Dihydroxanthyletin-Type Coumarins from Angelica decursiva with Protein Tyrosine Phosphatase 1B and α-Glucosidase Inhibitory Activities and Docking Studies of Their Molecular Mechanisms"

_antioxidants, 2021, doi:10.3390/antiox10020292_

Round 1
Reviewer 1 Report
The manuscript by Md Yousof Ali et al. follow very closely the work of the same authors in Chem. Biol. Interact.,252, 2016, 93-101 (ref. 31 of the manuscript).
Indeed, the work study some dihydroxanthyletin-type coumarins from Angelica Decursiva that are very similar to those studied in ref. 31 using completely analogous procedures and pursuing the same objective of being used to treat diabetes mellitus. The new compounds are derivatives of those already studied namely in ref. 31 and so the novelty of the work is very limited. In addition, the authors do not make a comparative approach between the compounds.
In conclusion the manuscript still contain material that deserves publication, but in order to be acceptable for publication should be strongly reformulated and shortened considering the work already published in the field.
Author Response
Dear Editor in Chief,
Thank you very much for your kind consideration and valuable comments from referee’s and editors’ side regarding the manuscript “Insulin–mimetic dihydroxanthyletin-type coumarins from Angelica decursiva with protein tyrosine phosphatase 1B and α-glucosidase inhibitory activities and docking studies of their molecular mechanisms”. I have corrected the manuscript according to the referee’s suggestions and sending herewith a copy of the revised manuscript for reconsideration. Each point raised by the reviewer was either justified or changed according to suggestions. All queries are answered to the point and revisions are marked as red colored font throughout the manuscript. Additionally, we have rephrased the manuscript according to the editor’s suggestions, marked as blue color font throughout the manuscript.
Response to Reviewers’ Comments
Ref: Antioxidants-1083019
Journal: Antioxidant
Reviewer #1
The manuscript by Md Yousof Ali et al. follow very closely the work of the same authors in Chem. Biol. Interact.,252, 2016, 93-101 (ref. 31 of the manuscript). Indeed, the work study some dihydroxanthyletin-type coumarins from Angelica decursiva that are very similar to those studied in ref. 31 using completely analogous procedures and pursuing the same objective of being used to treat diabetes mellitus. The new compounds are derivatives of those already studied namely in ref. 31 and so the novelty of the work is very limited. In addition, the authors do not make a comparative approach between the compounds.
In conclusion the manuscript still contain material that deserves publication, but in order to be acceptable for publication should be strongly reformulated and shortened considering the work already published in the field.
Reply 1: We greatly appreciate reviewer 1’s critical comments and suggestions. We are agreeing with the reviewer comments that we have previously reported the manuscript [ref-31], which is similar to current manuscript. But previous studies only focused on the scanning of PTP1B and α-glucosidase inhibitors via in vitro assays, and most of the compounds were minor compounds that present in Angelica decursiva. However, in the present study we have selected major dihydroxanthyletin-type coumarins from Angelica decursiva. Moreover, our current experimental strategies are different from previous study and this time we used cellular experiments using insulin resistant HepG2 cells and antioxidant mechanisms. As we know that the effective control of hyperglycemia is a prerequisite to control the incidence, progress, and severity of diabetes and single therapy eventually fails to control DM. Therefore, additional adjunct therapy like antioxidant mechanisms could be effective therapeutic options to regulate the DM. As DM is multifactorial diseases, therefore in the present study, we focused on the anti-diabetic potential of dihydroxanthyletin-type coumarins against multiple targets involved in the pathogenesis of DM. In addition, our isolated coumarins -(+)-trans-decursidinol, Pd-C-II, and Pd-C-III are rare type molecules that only exist in Angelica decursiva. Regarding reformulated and shortened issues, we have shortened our manuscript by rephrase of the whole manuscript. We hope our experimental tactics successfully addressed the main objective of the manuscript which will provide some necessary information for other interested researchers for further investigation.
Reviewer 2 Report
The manuscript describes a work aimed to to investigate the mechanism, at molecular level and with model cells, at the basis of fours natural coumarins from Angelica decursiva.
The manuscript is interesting and clearly written in all sections.
The state of the art is well documented and focuses on the topic precisely.
The aims are clearly stated.
The methodologies adopted are described with details and the general experimental is are definitively appropriate to achieve the aims.
The results are very well described and convincing. And are quite interesting and promising.
The discussion and conclusions are well addressed and supported by the experimental data and contextualized in the frame of the most updated knowledges on the topic.
I suggest very few minor revision.
Introduction: it would be interesting to have some hints on the possible presence of the studied compounds in other Angelica species or other plants.
Line 79. nodakenin in lowercase letters
line 170. "...20 μL of the sample dissolved in 10% DMSO was added to each..." is "20 μL of increasing amounts of sample dissolved in 10% DMSO was added to each..."?
Figure 2 and Figure 3. I would split the two figures in order to report separately the Dixon plots and the Lineweaver-Burk plot. In the current version the legends and captions are very difficult to read.
Figure 6. In the current representation the ligands are very difficult to identify.
Figure 7. The pictures appear of low quality and "stretched".
line 689. Please remove "Moreover". and correct to evidences.
line 693-695. I would add something like "could complement pharmacotherapy in the management of diabetes"
Author Response
Dear Editor in Chief,
Thank you very much for your kind consideration and valuable comments from referee’s and editors’ side regarding the manuscript “Insulin–mimetic dihydroxanthyletin-type coumarins from Angelica decursiva with protein tyrosine phosphatase 1B and α-glucosidase inhibitory activities and docking studies of their molecular mechanisms”. I have corrected the manuscript according to the referee’s suggestions and sending herewith a copy of the revised manuscript for reconsideration. Each point raised by the reviewer was either justified or changed according to suggestions. All queries are answered to the point and revisions are marked as red colored font throughout the manuscript. Additionally, we have rephrased the manuscript according to the editor’s suggestions, marked as blue color font throughout the manuscript.
Response to Reviewers’ Comments
Ref: Antioxidants-1083019
Journal: Antioxidant
Reviewer #2
- The manuscript describes a work aimed to investigate the mechanism, at molecular level and with model cells, at the basis of fours natural coumarins from Angelica decursiva. The manuscript is interesting and clearly written in all sections. The state of the art is well documented and focuses on the topic precisely. The aims are clearly stated. The methodologies adopted are described with details and the general experimental is are definitively appropriate to achieve the aims. The results are very well described and convincing. And are quite interesting and promising. The discussion and conclusions are well addressed and supported by the experimental data and contextualized in the frame of the most updated knowledges on the topic.
Reply 2-1: We greatly appreciate the reviewer’s valuable comments regarding our manuscript.
- Introduction: it would be interesting to have some hints on the possible presence of the studied compounds in other Angelica species or other plants.
Reply 2-2: Thank you very much for your valuable comments and suggestions regarding our manuscript. Only compound Pd-C-I has been identified from another Angelica species Peucedanum praeruptorum [37], otherwise the compounds -(+)-trans-decursidinol, Pd-C-II, and Pd-C-III has been reported for first in Angelica decursiva [38] not any plant species.
- Line 79. nodakenin in lowercase letters
Reply 2-3: Thank you very much for your valuable comments regarding our manuscript. We have changed the nodakenin lowercase letter marked as red color in line 78.
- line 170. "...20 μL of the sample dissolved in 10% DMSO was added to each..." is "20 μL of increasing amounts of sample dissolved in 10% DMSO was added to each..."?
Reply 2-4: Thank you very much for your valuable comments and suggestions regarding our manuscript. We have revised the sentence and marked it as red in line 170.
- Figure 2 and Figure 3. I would split the two figures in order to report separately the Dixon plots and the Lineweaver-Burk plot. In the current version the legends and captions are very difficult to read.
Reply 2-5: Thank you very much for your valuable suggestions regarding our manuscript. According to the reviewer suggestions, we have separated the figures 2 and 3. Please see the split figures marked as figure 2 and 3; figure 4 and 5 respectively.
- Figure 6. In the current representation the ligands are very difficult to identify.
Reply 2-6: Thank you very much for your valuable comments regarding our manuscript. According to the reviewer suggestions, we have improved the image quality and marked it as figure 8.
- Figure 7. The pictures appear of low quality and "stretched".
Reply 2-7: Thank you very much for your valuable comments regarding our manuscript. According to the reviewer suggestions, we have improved the image quality and marked it as figure 9.
- line 689. Please remove "Moreover". and correct to evidences.
Reply 2-8: Thank you very much for your valuable suggestions regarding our manuscript. According to the reviewer’s suggestions, we have removed the word “Moreover” and corrected the word “evidences”.
- line 693-695. I would add something like "could complement pharmacotherapy in the management of diabetes"
Reply 2-9: Thank you very much for your valuable comments regarding our manuscript. According to the reviewer’s suggestions, we have revised the sentence in the conclusion section marked as red.
Round 2
Reviewer 1 Report
The revised version of the manuscript was clarified and improved. Nevertheless, the authors do not follow the suggestion to compare these results with those presented before and do not reduce the length of the manuscript. I recommend the following:
- Reduction of the experimental procedures eliminating details described before in ref. 31.
- Reformulated Figure 2 to 5 to a maximum of 2 Figures. It is not necessary to present the Lineweaver-Burk and Dixon plots for all inhibitors and the two enzymes. Large amount of this material can be shown in Support Information (SI).
- Reducing the number of Figures of the docking part to 1/2.
Now concerning the comparison with the results published in ref. 31, I recommend the following:
- The inhibitory effect of the coumarins studied in this work must be compared with the values obtained in ref. 31. Namely the influence of and hydroxy group at position C-4’ must be compared considering the IC50 values found before for the 4’-Methoy Pd-C-1 with those determined here for Pd-C1 since both compounds differ only in the C-4’ substitution (methoxy and hydroxyl groups) and also the values obtained for Decursinol and Decursidinol. On the other hand, the results of Pd-CIII must be compared with the values obtained for Decursidin that are structurally very similar. Namely the authors should attempt to explain the reason for the IC50 index being practically equal for PTP1B but significantly different for the alpha-glucosidase. The authors should also give a justification for the differences in the IC50 values of the control compounds (Ursolic acid and Acarbose).
- The authors should compare and discuss the performance of all the compounds studied until now.
In conclusion, this work extends the work of ref. 31 by studying different compounds, although structurally very similar in vitro, and now also in HepG2 cell lines. If the authors consider the suggestions I will recommend the publication.
Author Response
Dear Editor in Chief,
Thank you very much for your kind consideration and valuable comments from referee’s and editors’ side regarding the manuscript “Insulin–mimetic dihydroxanthyletin-type coumarins from Angelica decursiva with protein tyrosine phosphatase 1B and α-glucosidase inhibitory activities and docking studies of their molecular mechanisms”. I have corrected the manuscript according to the referee’s suggestions and sending herewith a copy of the revised manuscript for reconsideration. Each point raised by the reviewer was either justified or changed according to suggestions. All queries are answered to the point and revisions are marked as red colored font throughout the manuscript. Additionally, we have rephrased the manuscript according to the editor’s suggestions, marked as blue color font throughout the manuscript.
Response to Reviewers’ Comments
Ref: Antioxidants-1083019
Journal: Antioxidant
Reviewer #1
The revised version of the manuscript was clarified and improved. Nevertheless, the authors do not follow the suggestion to compare these results with those presented before and do not reduce the length of the manuscript. I recommend the following:
General reply to reviewer 1: We greatly appreciates reviewer 1’s critical comments and suggestions.
- Reduction of the experimental procedures eliminating details described before in ref. 31.
Reply 1-1: We greatly appreciate the reviewer’s valuable comments regarding our manuscript. According to the reviewer suggestion, we have reduced the experimental procedure by deleting the information that is similar to our previous study [31]. Please see the revised information marked as red at page 4.
The following sentences have been deleted from the sub-heading 2.5 in the materials and methods section.
40 µL of PTP1B enzyme [0.5 units diluted with a PTP1B reaction buffer containing 50 mM citrate (pH 6.0), 0.1 M NaCl, 1 mM EDTA, and 1 mM DTT] were applied to each well of a 96-well plate (each with final volume of 100 µL), with or without a sample dissolved in 10% DMSO. The plate was pre-incubated for 10 min at 37°C, and then 50 µL of 2 mM pNPP was applied to the PTP1B reaction buffer. The reaction was terminated by the addition of 10 M NaOH after incubation at 37°C for 20 min. The nonenzymatic hydrolysis of 2 mM pNPP was corrected for the measured increase in absorbance at 405 nm obtained in the absence of PTP1B enzyme
The following sentences have been deleted from the sub-heading 2.6 in the materials and methods section.
A total of 60 µL of reaction mixture containing 20 µL of 100 mM phosphate buffer (pH 6.8), 20 µL of 2.5 mM pNPG, and 20 µL of increasing amounts of sample dissolved in 10% DMSO was added to each well, followed by 20 µL of α-glucosidase [0.2 U/mL in 10 mM phosphate buffer (pH 6.8)]. The plate was incubated for 15 min at 37°C and then applied 80 µL of 0.2 M sodium carbonate solution to avoid the reaction. The percentage of inhibition (%) was achieved using the same equation as in the PTP1B enzymatic assay.
The following sentences have been deleted from the sub-heading 2.7 in the materials and methods section.
95 µL BSA (0.4 mg protein/mL) and combined with 2.5 µL ONOO– (200 μM) were applied to different concentrations of coumarins dissolved in 10% DMSO. The sample was applied to the Bio-Rad gel buffer in a 1:1 ratio after incubation with shaking at 37°C for 20 min and boiled for 5 min to denature the proteins. In 80V for 30 min, 10% SDS-polyacrylamide mini-gel and 100V for 1 h were separated for the total protein equivalent, and then transferred to a PVDF membrane at 80V during 110 min, by means of a wet transfer system (Bio-Rad). The membrane was immediately put at room temperature for 1 h in a blocking solution (5% non-fat dry milk in TBS-Tween buffer (w/v), Bio-Rad TBS, and 0.1% Tween-20, pH 7.4). The membrane was washed three times (10 min each wash) in TBS-Tween buffer and incubated at 4°C overnight with a monoclonal anti-nitrotyrosine antibody (diluted 1:2,500 in TBS-Tween buffer with 5% non-fat dry milk). After three additional washes in the TBST buffer (10 min and 5 min), the membrane was incubated for 1 h at room temperature with horseradish peroxidase-conjugated sheep anti-mouse secondary antibody diluted 1:2,000 in TBST buffer. According to the manufacturer’s instructions, antibody labeling was visualized After three washes in TBST buffer using the Super signal West Pico Chemiluminescent substrate (Pierce, Rockford, IL, USA). X-ray film (Kodak, Rochester, NY, USA) was then exposed to the membrane. For molecular weight determination, pre-stained blue protein markers were used.
- Reformulated Figure 2 to 5 to a maximum of 2 Figures. It is not necessary to present the Lineweaver-Burk and Dixon plots for all inhibitors and the two enzymes. Large amount of this material can be shown in Support Information (SI).
Reply 1-2: Thank you very much for your valuable comments and suggestions regarding our manuscript. According to the reviewer suggestions, we have combined figures 2 to 5 to make 2 figures that are shown in supporting information Figure S1 and Figure S2. Please see the supporting information in the supplementary file.
- Reducing the number of Figures of the docking part to 1/2.
Reply 2-3: Thank you very much for your valuable comments regarding our manuscript. According to the reviewer suggestions, we have reduced the docking figures 6 and 8. Please see figure 6 and figure 8 in the supplementary file, marked as Figure S3 and Figure S4.
- Now concerning the comparison with the results published in ref. 31, I recommend the following:
The inhibitory effect of the coumarins studied in this work must be compared with the values obtained in ref. 31. Namely the influence of and hydroxy group at position C-4’ must be compared considering the IC50 values found before for the 4’-Methoy Pd-C-1 with those determined here for Pd-C1 since both compounds differ only in the C-4’ substitution (methoxy and hydroxyl groups) and also the values obtained for decursinol and decursidinol. On the other hand, the results of Pd-C III must be compared with the values obtained for decursidin that are structurally very similar. Namely the authors should attempt to explain the reason for the IC50 index being practically equal for PTP1B but significantly different for the alpha-glucosidase. The authors should also give a justification for the differences in the IC50 values of the control compounds (Ursolic acid and Acarbose).
Reply 1-4: Thank you very much for your valuable comments and suggestions regarding our manuscript. According to the reviewer suggestions, we have included the information about the previous report [31], and current research, that marked as red color at page-7 and in line- 320-334. Moreover, we also compared our results with positive controls urosolic acid and acarbose, please see at page 7, in line 301 and 305. Regarding the issue of IC50 values difference of urosolic acid and acarbose, generally these two compounds IC50 values range between 2.50-10 µM for ursolic acid and 100-500 µM for acarbose. Therefore, the IC50 values of urosolic acid and acarbose were slightly different due to the experimental parameter changes.
- The authors should compare and discuss the performance of all the compounds studied until now.
In conclusion, this work extends the work of ref. 31 by studying different compounds, although structurally very similar in vitro, and now also in HepG2 cell lines. If the authors, consider the suggestions I will recommend the publication.
Reply 1-5: Thank you very much for your valuable suggestions regarding our manuscript. According to the reviewer suggestions, we have included the information about different groups of compounds and compared between them. Please see the revised information at page-20, line 617-627, marked as red color.
Round 3
Reviewer 1 Report
The authors reformulate the manuscript according to the referre sugestions, In my opinion the manuscript was improved and I recommend publication.